# Durability of Shield Effective Polyester Cotton Fabric with Integrated Stainless Steel Threads in Processes of Dry and Wet Cleaning

**DOI:** 10.3390/polym15030633

**Published:** 2023-01-26

**Authors:** Tanja Pušić, Bosiljka Šaravanja, Anica Hursa Šajatović, Krešimir Malarić

**Affiliations:** 1Faculty of Textile Technology, University of Zagreb, Prilaz baruna Filipovića 28a, 10000 Zagreb, Croatia; 2Faculty of Electrical Engineering and Computing, University of Zagreb, Unska 3, 10000 Zagreb, Croatia

**Keywords:** fabric with integrated metal threads, shield effectiveness, dry cleaning, wet cleaning

## Abstract

Protective properties against electromagnetic radiation represent the essential value of textiles for protective clothing worn in hazardous working conditions. To ensure that protective clothing lasts for as long as feasible, care processes must be optimized, especially since protective clothing is subjected to repeated cycles due to soiling. Improved formulations of special detergents and agents with high-performance finishing agents used in care processes can contribute to a longer life of protective clothing. In this paper, the shield effectiveness of a cotton polyester fabric with integrated metal threads in the warp and weft directions at frequencies of 0.9 GHz, 1.8 GHz, 2.1 GHz, and 2.4 GHz is compared to that of its environmentally friendly alternative, wet cleaning processes. The research is carried out by varying process parameters: mechanics, chemical action, and cycle number. The results proved the drop of shield properties of polyester/cotton fabric with integrated stainless steel threads depending on frequencies and the number of professional cleaning cycles. The worst shield effectiveness of the tested fabric was obtained after 10 cycles of dry cleaning, wherein the degree of protection at frequency 2.4 GHz is reduced by 6.89 dB. According to the obtained results, the level of functional properties was better preserved in the wet cleaning process, which additionally has better ecological premises compared to dry cleaning process.

## 1. Introduction

Today, there is a rising concern among the general population regarding the electromagnetic radiation, whether justified or not. In some cases, the general public as well as industry need to protect vulnerable members of their families (such as children in cradles), or sensitive electronic equipment in industrial facilities. In such situations, an effective and replaceable electromagnetic shield is desirable. Textile materials combined with metallic threads could provide an adequate solution for the situations mentioned above. Textile materials, since they are lightweight and flexible, can operate as a shield against dangerous electromagnetic radiations (EMR) [1,2,3]. Textile surfaces can be converted into EMR shields by incorporating metallic components into the textile skeleton. Shield-acting fabrics counteract EMR by reflection [4,5,6,7]. 

This mechanism’s shielding safeguards both humans and technology. It is important to carefully design textiles suitable to mitigate external hazards [8,9]. Shield-textile structures should be designed to maximize comfort, wearability, breathability, and durability as extra features for clothing. Shield-acting textiles created by incorporating metallic threads into woven and knitted fabrics were tested for washing and drying. The shield effect of such structures is reduced by the impact of chemical, mechanical, and thermal activities that cause degradation and migration of metal fibers from the threads [10,11,12].

A washing process with a particular functional detergent is recommended on the care label for the preservation of the shield effectiveness (SE) qualities of functional fabrics coated with various metals [13,14]. Under cyclic dry and wet cleaning operations, the original SE characteristics of polyamide-knitted fabric made of silver-coated thread were altered. The scanning electron microscope examination of the surface revealed that the silver coating on the polyamide yarn had been peeled. The level of degradation of the functional silver coating after wet washing was higher than after dry cleaning [15].

If functional fabrics are treated environmentally, with friendly care and procedures, they can earn the sustainable label. Care procedures are adapted to the development of textile materials and technologies while adhering to environmental rules. Greener detergents, eco-friendly dry cleaning solvents, and machine adaption for care operations are the foundations of the progress, which calls for interdisciplinary collaboration [16,17]. 

It is important to point out the key differences between the washing process and professional care processes (dry cleaning and wet cleaning). All processes are conceptually set according to Sinner’s circle through the interaction of four primary factors: chemistry (selection of agent and solvent), mechanics (stain removal by centrifugal forces or friction), temperature, and time (duration of a process). During textile care, it is critical to consider degrading influences. Mechanical activity through centrifugal and friction impacts is smaller in wet and dry cleaning compared to washing. 

The dry cleaning process is carried out in organic solvents with the addition of boosters. Solvents differ in terms of volatility, toxicity, and cleansing impact, which is commonly stated in terms of KBV, or Kauri-butanol value [18].

Wet cleaning is a low-temperature procedure characterized by very gentle mechanics and short cycles. The process is carried out in water using environmentally friendly mild detergents and special additives. Each of the procedures listed has advantages and limitations that must be considered from a technological and environmental s view [16]. 

The benefits of the wet cleaning technique include its suitability for a wide range of textiles, the absence of toxic chemicals, and the reduction in air and water pollution. Water-based stains are effectively cleaned, and 50% less energy is consumed compared to dry cleaning which is characterized by high enthalpies [16]. 

The inability to remove severe soiling (fats, wax, and oil), shrinkage, surface alterations, felting, loss of gloss, and discoloration of textiles are the limits of wet cleaning as compared to dry cleaning. The unfavorable qualities stated above can be mitigated by using functional additives that are detergent compatible.

Special finishing agents should not degrade textile characteristics. Softeners and related finishing agents are used in synthetic textiles to minimize or eliminate static charge [19]. The formation of a lubricating film on the surface improves softness, fluffiness, anti-wrinkle, and easy iron effects. 

Previous research found that cyclic washing of electromagnetic shielding fabric (EMS)—polyester/cotton fabric with integrated stainless steel threads (PES/CO/SS)—with specifically formulated powder and liquid detergent had an effect on the loss of protective characteristics. If the electromagnetic shield effect (EMSE) is monitored at higher frequencies, the composition of the powder and liquid detergents is appropriate [20]. 

We have investigated in our research the effect of cleaning (both dry and wet) to analyze how it affects the electric properties such as shielding effectiveness (SE) of polyester cotton fabric with integrated stainless steel threads. The dry and wet cleaning durability of the EMS fabric made of polyester/cotton fabric with stainless steel threads in the warp and weft directions regarding the SE on different frequencies (0.9 GHz, 1.8 GHz, 2.1. GHz and 2.4. GHz) which are widely used for mobile telephony, WLAN, Bluetooth and microwave oven. Shielding properties of the fabric were measured before and after professional care processes, including dry cleaning and wet cleaning in two variations at frequencies of 0.9 GHz, 1.8 GHz, 2.1 GHz, and 2.4 GHz. The influence of process parameters of dry and wet cleaning on the mechanical damage of the fabric was additionally analyzed using microscopic examination of the surface [20].

## 2. Materials and Methods

### 2.1. Textile Material

An electromagnetic shielding fabric used for the production of protective clothing was designed and produced using the weaving machine OMNIplus 800, Picanol, Ieper, Belgium in the Čateks factory, Čakovec, Croatia. The fabric is made of 49% polyester, 48% cotton and 3% stainless steel (SS), and woven in twill K2/2 with a mass per unit area of 249 g/m^2^. Since the stainless steel is an inexpensive and has good EM-reflection qualities, SS can be good EM-shielding material [21]. The thread with SS is integrated in the warp and weft directions, with a density in the warp direction of 38 plus 1 and 20 plus 1 in the weft direction.

### 2.2. Procedures

In comparison to the pristine fabric, the protective properties of this functional fabric were measured after the first, third, fifth, seventh, and tenth cycles of dry and wet cleaning.

#### 2.2.1. Dry Cleaning

The dry cleaning process was conducted in a dry cleaning machine, Renzacci SpA Industria Lavatrici, Città di Castello, Itally, using a two-bath procedure with perchloroethylene as the solvent (P) at 20 °C during 10 min, as it is presented in Table 1.

In the first bath, a booster in concentration 2% (per mass of textiles) was applied to strengthen the cleaning impact of perchloroethylene and the mitigation of the antistatic effects of the fabric in pre-cleaning, while the booster in the second bath is not added. Fabrics were subjected to 10 cycles of dry cleaning and subsequent drying at 60 °C during 30 min including reverse rotations (5/5). Fabrics were ironed at 110 °C after the first, third, fifth, seventh, and tenth dry cleaning cycles.

#### 2.2.2. Wet Cleaning

The wet cleaning procedure in the Grandimpianti GWH 135/14 kg machine, Sospirolo, Italy, was carried out in two versions: wet cleaning with detergent, and wet cleaning with detergent and a high-performance finishing agent. 

The detergent specified in Table 2 and the high-performance finishing agent specified were employed in both the W and W* wet cleaning procedures. The highly efficient liquid color detergent is aimed for professional treatment of washable outer clothes, sports, and work wear (PPE, uniforms, corporate identity fabric).

Detergent ingredients and their content in percentage are marked according to Regulation (EC) No 648/2004 of the European Parliament and of the Council.

The wet cleaning detergent contains anionic and non-ionic surfactants, solubility promoter, pH buffer, alkali, preservatives, and fragrances. 

High-performance finishing agents have a beneficial impact on drying behavior, the ability to reduce final finishing/ironing, and the ability to preserve the natural qualities of the fibers. 

The agent is based on the polyvinyl acetate (>50.0%) as a water-soluble polymer aimed for improvement of touch, softness, and antistatic properties. The modified methylhydroxycellulose as a thickening agent was a low-concentration additive in this aqueous formulation. PES/CO/SS fabric is wet cleaned with detergent and wet cleaned with detergent and finishing agent according to the protocols shown in Table 3 and Table 4.

The wet cleaning of EMS functional fabric with integrated metal threads using detergent and high performance finishing agent (W*) is carried out in accordance with the specifications shown in Table 4.

Both processes (W and W*) are low temperature and performed at 20 °C during 22 min. Fabrics are dried in the Grandimpianti GD 350/14 kg dryer, Sospirolo, Italy at 50 °C for 2 min after the first, third, fifth, seventh, and tenth wet cleaning cycles, and then in the air.

### 2.3. Measurement of EM Shielding Effectiveness 

The effect of dry and wet cleaning cycles on the shielding capabilities of the fabric with metal treads was studied using a method developed at the University of Zagreb’s Faculty of Electrical Engineering and Computing’s Microwave Laboratory in the Department of Wireless Communications. Figure 1 shows the measurement equipment (Selective Radiation Meter SRM—3000, Narda Safety Test Solutions GmbH, Pfullingen, Germany; High performance—HP 8350 B signal generator, Agilent, US; IEV horn antenna—A12 type, Telecommunications industry, Ljubljana, Slovenia and the wooden frame for the functional fabric (1 m × 1 m) was used for EM field intensity measurements according to the protocols of the ASTM D-4935:2018, IEEE Std 299-2006 and MIL STD 285 [22,23,24].

The ratio between the EM field intensity (*E_0_*) measured without the functional fabric and the EM field intensity (*E_1_*) with the functional fabric positioned between the radiation source and the measuring instrument was used to calculate the EM protection factor. Shielding effectiveness can be calculated according to Equation (1) [26,27]:SE = 20 log(*E_1_/E_2_*)(1)
where *E_1_* is electric field measured without the shield and *E_2_* is electric field measured with the shield.

At frequencies of 1.8 GHz, 2.1 GHz, and 2.4 GHz, the SE characteristics of functional protective textiles were measured before and after 10 cycles of dry and wet cleaning. 

### 2.4. Surface Appearance

The surface of pristine, dry cleaned, and wet cleaned fabrics was observed under magnification of 235× using a digital microscope DinoLite, Premier IDCP B.V., Almere, The Netherlands.

## 3. Results and Discussion

The impact of individual cycles (1st, 3rd, 5th, 7th, and 10th) of professional care on SE of PES/CO/SS fabric from face and back side at frequencies 0.9 GHz, 1.8 GHz, 2.1 GHz, and 2.4 GHz was analyzed.

The shielding effect (SE) of the pristine sample at a frequency of 0.9 GHz is 7.20 dB. By increasing the dry cleaning cycles, SE drops by about 1 dB and after 10 cycles reaches 3.26 dB, which makes a difference loss of 3.94 dB compared to the pristine one.

The pristine sample has the best SE at the frequency of 2.4 GHz that is 17.60 dB. It gradually decreases and after 10 dry cleaning cycles, the SE is 10.71 dB, indicating that the degree of protection has been reduced by 6.89 dB, Figure 2.

Figure 3 shows the SE of a fabric (back side) prior to and following dry cleaning cycles.

When treated with dry cleaning, the back side of the PES/CO/SS fabric performs somewhat poorer than the face side of the fabric, Figure 3. At a frequency of 0.9 and 2.4 GHz, the SE loss is almost the same. At 0.9 GHz, the loss is 4.06 dB, while at 2.4 GHz, the loss is 7.45 dB. At frequencies 1.8 GHz and 2.1 GHz, there are minor differences between the face and the back.

Figure 4 and Figure 5 show the protection efficiency of a fabric (face and back) before and after a wet cleaning cycle (W).

The difference in the degree of SE is slightly reduced at all frequencies, according to the SE results of the fabric sample with stainless steel (face) after wet cleaning with detergent (W) compared to pristine sample (0). The loss SE of the 10 times treated sample (W-10) at 0.9 GHz is 3.44 dB. At 1.8 GHz, the loss is 4.2 dB, whereas at 2.1 GHz, the loss is 5.6 dB. The loss in relation to the starting value at 2.4 GHz is 6.87 dB.

According to the results of the stainless steel back side of the fabric, the SE of the pristine sample is negligibly lower than of the face side at all frequencies. The SE variations following wet cleaning cycles vs. the starting values are almost identical to those for the fabric’s stainless steel face side.

The SE of a fabric sample with stainless steel (face side) is shown in Figure 6 both before and after the wet cleaning treatment cycle (W*).

The fabric sample after a wet cleaning process with detergent and a high performance functional finishing agent (W*) is designed to keep materials’ functional protective characteristics intact.

Compared to wet cleaned samples, the reduction in SE of the wet cleaned sample (W*) is less at all frequencies (W). The results are better than expected when compared to the fabric sample that was not treated with this chemical (W).

Figure 7 shows the SE of a stainless steel fabric sample (back side) before and after a wet cleaning cycle with a high-performance finishing agent (W*).

The differences in SE findings between the face and back sides are still negligible. The decrease in SE at frequencies 0.9 and 2.4 GHz is less linear than at frequencies 1.8 and 2.1 GHz.

The results showed that cleaning has an effect on SE of textile material with metallic threads which depends on the frequency and the number of cleaning cycles.

### 3.1. Comparative Analysis

Table 5 and Table 6 provide a comparison of the SE values obtained after 10 cycles of dry and wet cleaning of the PES/CO/SS fabric measured from the face and back side.

Table 5 shows that 10 cycles of dry cleaning had a greater effect on the decline in protective characteristics than 10 cycles of wet cleaning. The results at all frequencies confirm the beneficial influence of the high performance finishing agent (W*).

When measuring the opposite side of the studied PES/CO/SS fabric, a similar pattern was found.

The loss of SE occurred due to mechanical strain present during tumbling of textiles and since the SS threads were broken during, the process and this resulted with less metallic rupture which in turn resulted with loss of SE.

The SE in all cases was lost for the frequency of 0.9 GHz because the mesh characteristics of inox wires loss such to give the best protection for frequency of 2.4. GHz. The neighboring frequencies of 2.1. GHz and 1.8 GHz also give better protection since they are very close regarding to frequency. The higher frequency such as the example 3.5 GHz would give similar results as the frequency of 0.9 GHz due to dimensions of mesh and resonance.

### 3.2. The Surface Appearance

Table 7 shows the surface of PES/CO/SS fabric before and after 10 cycles of dry washing (P-10), wet cleaning (W-10), and wet cleaning with the addition of HP finishing agent (W*-10).

The surface of the PES/CO/SS fabric has been modified after 10 cycles of professional care in comparison to the pristine fabric. As mechanical agitation in dry and wet cleaning is gentle, the most prominent factor for surface changes can be attributed to the chemical action, which is different in tested processes. Perchloroethylene (P) is used in the dry cleaning process together with a booster, specially formulated detergent (W) is used in the wet cleaning process, whereas an HP finishing agent is added to the detergent in the wet cleaning process (W*).

Due to the percentage of cotton in the blend, the change in fibrillation is chosen as a surface characteristic. It is clear that 10 dry cleaning cycles resulted in a low degree of fibrillation due to the booster employed in the pre-cleaning phase. This difference is also noticeable after 10 cycles of wet cleaning. The lubricating effect of the HP finishing agent is obvious, and the fibrillation level is lower than in the wet cleaning with detergent process.

## 4. Conclusions

The effects of professional care on the SE characteristics of functional fabric made of PES/CO with SS thread are being researched. SE measurements at 0.9 GHz, 1.8 GHz, 2.1 GHz, and 2.4 GHz demonstrate the fabric’s protective qualities with CO/SS threads in the warp and weft directions. The results suggest that particular cycles of professional care had an effect on the loss of protective capabilities. It has been demonstrated that as the number of professional cleaning cycles increases, the SE properties of PES/CO/SS fabric diminish.

The intensity of changes at all frequencies is determined by the professional care procedure parameters. The dry cleaning process, despite the short duration, had the greatest impact on the loss of SE. The loss of SE properties can be attributed to the action of the solvent perchloroethylene with the addition of a booster and due to mechanical strain present during tumbling of textiles, on the stainless steel threads integrated in polyester/cotton fabric.

The wet cleaning process was carried out by a chemical change in the Sinner’s cycle. It has been demonstrated that the high performance finishing agent has a protective effect when compared to a specifically developed wet cleaning detergent. After 10 wet cleaning cycles, digital microscopic images verified its good effect on the surface. The results demonstrated that functionality and sustainability of specially designed functional fabrics with stainless steel threads is more permanent and stable in wet cleaning with detergent and high-performance finishing agent than in dry cleaning process. Future research will focus on the wet cleaning process, where different high-performance finishing agents will be employed in the concentration range. Regarding further experiments, the authors will also conduct their research on 5G frequencies such as 3.6 GHz, 3.7 GHz, 4.7 GHz, and beyond.

## Figures and Tables

**Figure 1 polymers-15-00633-f001:**
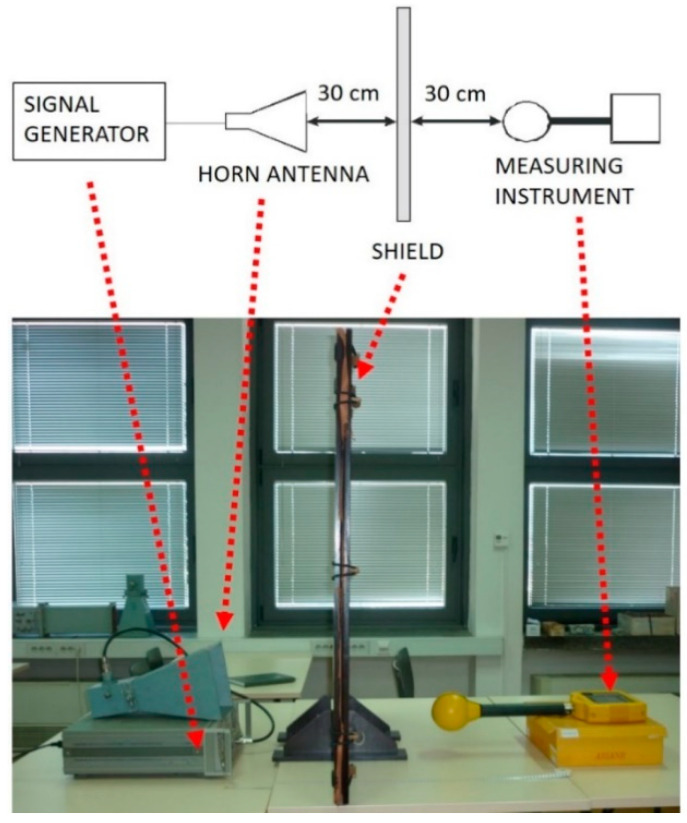
Measurement set up [25].

**Figure 2 polymers-15-00633-f002:**
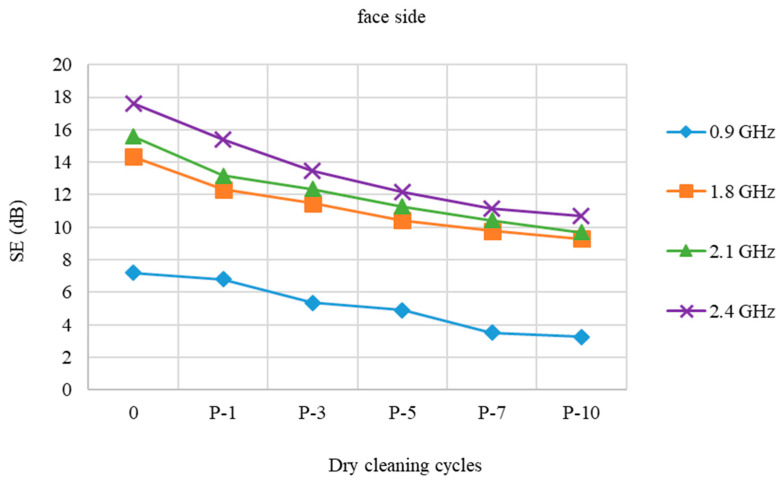
Shield effectiveness of PES/CO/SS fabric (face side) before and after 1st, 3rd, 5th, 7th, and 10th dry cleaning cycle (P) at frequencies 0.9 GHz, 1.8 GHz, 2.1 GHz, and 2.4GHz.

**Figure 3 polymers-15-00633-f003:**
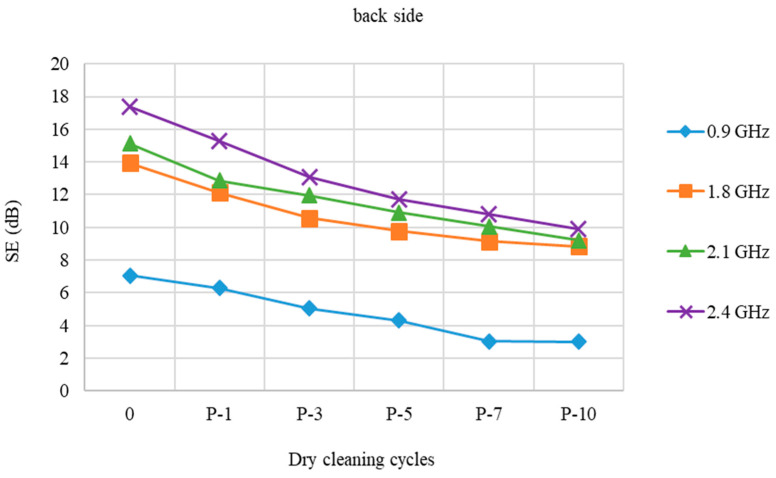
Shield effectiveness (SE) of PES/CO/SS fabric (back side) before and after 1st, 3rd, 5th, 7th, and 10th dry cleaning cycle (P) at frequencies 0.9 GHz, 1.8 GHz, 2.1 GHz, and 2.4 GHz.

**Figure 4 polymers-15-00633-f004:**
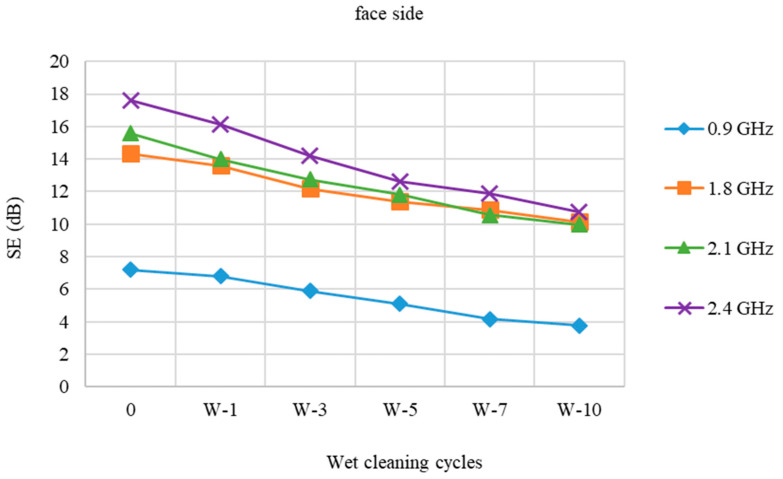
Shield effectiveness of PES/CO/SS fabric (face side) before and after 1st, 3rd, 5th, 7th, and 10th wet cleaning cycles (W) at frequencies 0.9 GHz, 1.8 GHz, 2.1 GHz, and 2.4 GHz.

**Figure 5 polymers-15-00633-f005:**
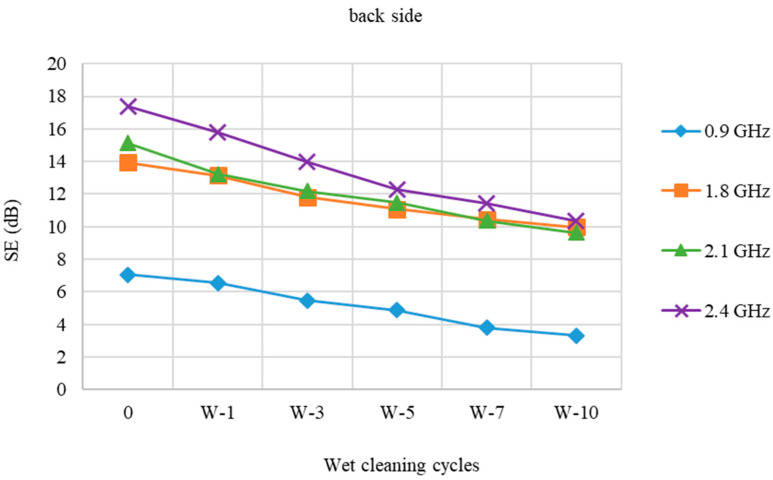
Shield effectiveness of PES/CO/SS fabric (back side) before and after 1st, 3rd, 5th, 7th, and 10th wet cleaning cycles (W) at frequencies 0.9 GHz, 1.8 GHz, 2.1 GHz, and 2.4 GHz.

**Figure 6 polymers-15-00633-f006:**
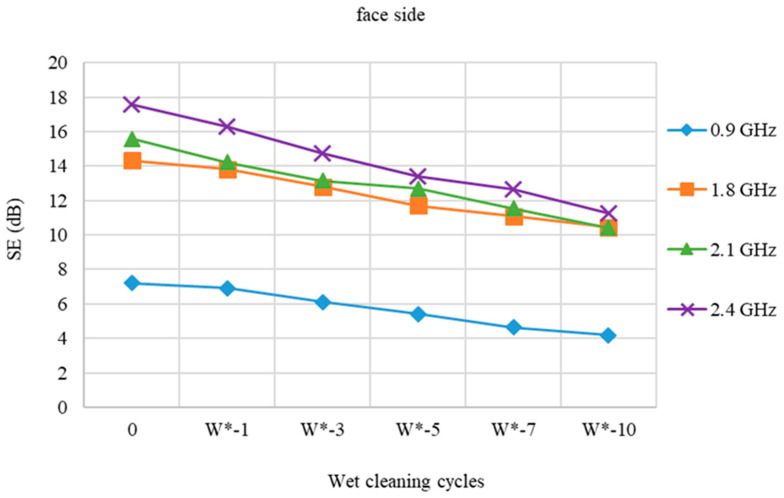
Shield effectiveness of PES/CO/SS fabric (face side) before and after 1st, 3rd, 5th, 7th, and 10th wet cleaning cycles (W*) at frequencies 0.9 GHz, 1.8 GHz, 2.1 GHz, and 2.4 GHz.

**Figure 7 polymers-15-00633-f007:**
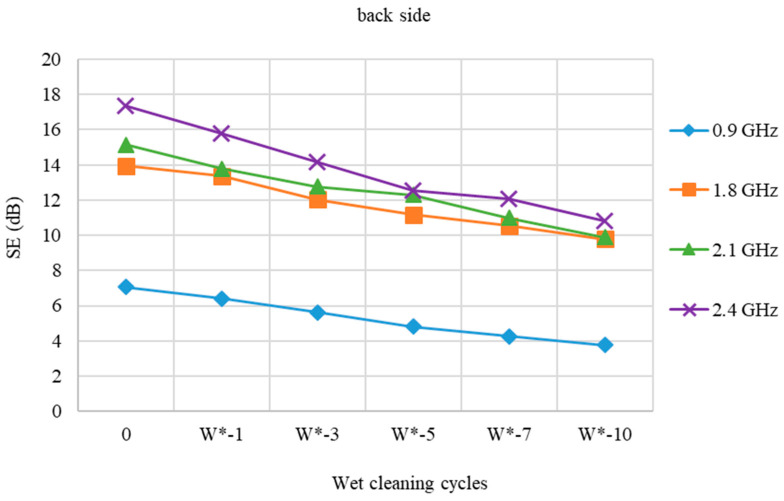
Shield effectiveness of PES/CO/SS fabric (back side) before and after 1st, 3rd, 5th, 7th, and 10th wet cleaning cycles (W*) at frequencies 0.9 GHz, 1.8 GHz, 2.1 GHz, and 2.4 GHz.

**Table 1 polymers-15-00633-t001:** Dry cleaning process.

Parameters	1st Bath	2nd Bath
	Pre-Cleaning	Cleaning
Temperature (°C)	20	20
Time (min)	4	6
Number of revolutions (min^−1^)	300	360
Volume of P (l)	20	40
Booster (%)	2	-
Bath ratio (BR)	1:2	1:4

**Table 2 polymers-15-00633-t002:** Composition of wet cleaning detergent.

Ingredient	w(%)
Isotridecanol, ethoxylated (5-20 EO *)	≥1.0, <10.0
Benzenesulfonic acid, alkyl derivates, sodium salts	≥1.0, <10.0
Alcohols, C10-12, ethoxylated propoxylated	≥1.0, <10.0
fatty alcohol alkoxylate	≥1.0, <10.0
2-Phenoxyethanol	≥1.0, <10.0
Sodium cumenesulphonate	≥1.0, <10.0
Benzyl alcohol	≥1.0, <10.0
Potassium hydroxide	≥0.1, <1.0
Orange, sweet, extract	≥ 0.1, <1.0
(2-Methoxymethylethoxy)-propanol (mixed isomers)	≥1.0, <10.0

* EO—etoxy groups.

**Table 3 polymers-15-00633-t003:** Wet cleaning (W) of PES/CO/SS fabric.

Step	Action (Run/Stop)	Time (min)	T (°C)	BR	Agents	Dosage (ml/kg)
Main wash	5/25	10	25	1:3.5	D	7
Extraction	Medium (450)	1	-	-	-	-
Rinse	5/25	3	cold	1:3.5		
Rinse	5/25	3	cold	1:3.5	-	-
Extraction	Delicate	5	-	-	-	-

**Table 4 polymers-15-00633-t004:** Wet cleaning (W*) of PES/CO/SS fabric.

Step	Action (Run/Stop)	Time (min)	T (°C)	BR	Agents	Dosage (ml/kg)
Main wash	5/25	10	25	1:3.5	D	7
Extraction	Medium (450)	1	-	-	-	-
Rinse	5/25	3	cold	1:3.5	F	6
Rinse	5/25	3	cold	1:3.5	-	-
Extraction	Delicate	5	-	-	-	-

**Table 5 polymers-15-00633-t005:** Shield effectiveness of PES/CO/SS fabric (face side) before and after 10 cycles of professional care (P-10, W-10, W*-10) at the frequencies 0.9 GHz, 1.8 GHz, 2.1 GHz, and 2.4 GHz.

**Fabric**	** *f* ** **(GHz)**			
	0.9	1.8	2.1	2.4
	**SE (dB)**			
Pristine	7.20	14.33	15.58	17.60
P-10	3.26	9.30	9.69	10.71
W-10	3.76	10.13	9.98	10.73
W*-10	4.18	10.44	10.41	11.29

**Table 6 polymers-15-00633-t006:** Shield effectiveness of PES/CO/SS fabric (back side) before and after 10 cycles of professional care (P-10, W-10, W*-10) at the frequencies 0.9 GHz, 1.8 GHz, 2.1 GHz, and 2.4 GHz.

**Fabric**	** *f* ** **(GHz)**			
	0.9	1.8	2.1	2.4
	**SE (dB)**			
Pristine	7.05	13.95	15.14	17.36
P-10	2.99	8.84	9.21	9.91
W-10	3.32	9.97	9.63	10.34
W*-10	3.76	9.79	9.88	10.84

**Table 7 polymers-15-00633-t007:** Digital micrographs of PES/CO/SS fabric before and after 10 cycles of dry cleaning (P-10), wet cleaning (W-10 and W*-10) with macro-sized fragments.

Pristine	P-10	W-10	W*-10
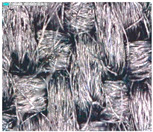	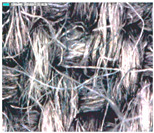	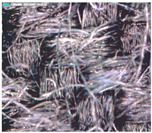	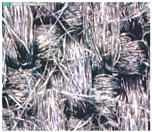
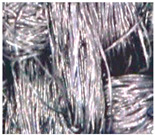	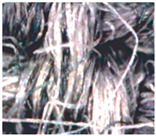	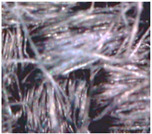	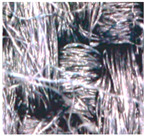

## Data Availability

The data presented in this study are available on request from the corresponding author.

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
