# Peer review of "Durability of Shield Effective Polyester Cotton Fabric with Integrated Stainless Steel Threads in Processes of Dry and Wet Cleaning"

_polymers, 2023, doi:10.3390/polym15030633_

Round 1

Reviewer 1 Report

In the Introduction part of the manuscript scientific problem could be presented and discussed in more specific manner. Authors should stated why this scientific problem is so important. The aim of the study should be emphasized in more specific manner, because in my opinion this part is not appropriately described. In this part authors may also add some information about the possible results and what they could possibly mean.

Research hypothesis should be specified  and connected with research problem. 

In methodological part I can not find statistical analysis. Authors should correlate obtained results. 

Conclusion presented should be more clear and providing inspiration for further research.

In generał presented manuscript is interesting and worth publishing after some slight changes which I mentioned above.

Author Response

Dear Reviewer

Thank You very much for comments and suggestions for improvement of a manuscript

Kind regards

Authors

Reviewer 2 Report

The topic is current since the effects of the growing use of EMR can be more discussed by the research community. The article is well-balanced and presents meaningful results. However, some minor changes can be made to improve it. In addition, I recommend a revision of the title, because both “functionality” and “sustainability” evaluation are not evaluated by the work carried out. Finally, a clear distinction and improvements in the author’s previously published work must be stated in the document (ref. 20).

I think that is not clear to all readers the need for protective clothes against EMR. I suggest that the authors elaborate on the “hazardous working conditions” stated in the abstract.

A short paragraph can be added to the introduction to justify the use of the three selected values of radio frequency (line 85).

I suggest removing the parenthesis in (SS) in lines 95-96.

Please avoid the repetition of abbreviations, such as “SE” in lines 81 and 98, (D) in lines, 116, 117, and 121, (HP) in lines 121 and 132, … (W) … (P) …, and so on. Please double-check the need for so many abbreviations and avoid their repetition.

Please add the model of the washing machine.

The meaning of “30 (5/5)” in Table 1 must be provided in the text.

Table 2 can be removed. W and W* can be defined in lines 116 and 117.

The applied Regulation referred to in lines 125 and 126 must be explicit.

Please clarify the composition of the wet cleaning detergent in table 3. Besides the ranges being large, the ranges given for potassium hydroxide and orange extract are unclear. Moreover, the meaning of (5-20 EO) must be provided, perhaps as a table footnote.

Like Table 3, Table 4 can be replaced by describing the ingredients in the main text.

Are the “Drain” lines relevant in Tables 5 and 6?

Section 3 presents results for 0.9 GHz, but previous sections only refer to 1.8 to 2.4 GHz.

Please check Fig. 3 and 4. They seem to be almost identical. Moreover, the results presented in Figs. 1 to 7 are all very similar. I suggest replacing them all with a Figure presenting all results for the sake of clarity.

Is line 226 a subsection? Why not be numbered accordingly? The same remark is for line 249.

I suggest adding to the conclusion sections a brief recommendation for future works.

It should be placed in a space between numbers and ºC.

Finally, the text must be checked for minor typo errors.

Author Response

(The authors gave the same response as above.)
